# Hierarchical Deep LSTM for Fault Detection and Diagnosis for a Chemical Process

**Piyush Agarwal** **, Jorge Ivan Mireles Gonzalez, Ali Elkamel and Hector Budman ***

Department of Chemical Engineering, University of Waterloo, Waterloo, ON N2L 3G1, Canada
* Correspondence: hbudman@uwaterloo.ca

**Abstract:** A hierarchical structure based on a Deep LSTM Supervised Autoencoder Neural Network (Deep LSTM-SAE NN) is presented for the detection and classification of faults in industrial plants. The proposed methodology has the ability to classify incipient faults that are difficult to detect and diagnose with traditional and many recent methods. Faults are grouped into different subsets according to the degree of difficulty to classify them accurately in the proposed hierarchical structure. External pseudo-random binary signals (PRBS) are injected in the system to enhance the identification of incipient faults. The approach is illustrated on the benchmark process (Tennessee Eastman Process) in order to compare across different methodologies. The efficacy of the proposed method is shown by a comprehensive comparison between many recent and traditional fault detection and diagnosis methods in the literature for Tennessee Eastman Process. The proposed work results in significant improvements in the classification of faults over both multivariate linear model-based strategies and non-hierarchical nonlinear model-based strategies.

**Keywords:** fault detection and diagnosis; statistical process monitoring (SPC); classification; autoencoders; deep learning; Tennessee Eastman Process; LSTM; incipient faults

## 1. Introduction

The faults in a chemical plant often propagate along the process, significantly impacting the profit of chemical plants. Hence, it is imperative to detect them soon upon their occurrence. The operation of industrial plants employs sensors and control loops to mitigate the economic losses resulting from these faults. However, in the presence of process faults and manipulated variable constraints, these control schemes are not sufficiently resilient to avoid abnormal operation [1,2]. Thus, process faults must be diagnosed and addressed by implementing a suitable corrective measure.

A typical process monitoring system consists of two parts: fault detection and diagnosis methodology. The objective of a fault detection system is to make a binary decision whether the current state of the process is in a normal or faulty operation region. Once an abnormal operation is detected, the fault diagnosis system is used to infer the type of fault or identify the root cause of the process fault. In the current study, we perform both detection and classification with a single algorithm by considering the normal operation condition as an additional fault class to be identified in the classification step.

Process monitoring schemes rely on estimated process models using historical data to infer faults. Based on the type of model, the methodologies are divided into two main approaches: mechanistic model-based (e.g., using first principles models) and data-driven model-based approaches [1]. Data-driven models for FDD, such as the one used in the current study, are based on a comparison between different sensor measurements under normal operation versus faulty operation [3–8]. Within the class of data-driven approaches, several reported algorithms are based on multivariate statistical methods such as Principal Component Analysis (PCA) [9–12] or its dynamic version such as Dynamic Principal Component Analysis (DPCA) [1,10,13–15]. These methods assume process behavior is

linear. However, most chemical processes are inherently non-linear in nature. Thus, non-linear modeling techniques such as Deep Neural Networks (DNNs) are employed in the current work. In the last decade, a new generation of Deep Neural Networks (DNNs) algorithms has emerged that capitalizes both the significant increase in computational power and novel algorithmic developments that facilitate the training and calibration of these networks. The use of these algorithms for fault detection in the process industry has recently received increased attention. However, despite the improvements in detection accuracy obtained with these techniques, some faults are still difficult to detect and diagnose (incipient faults). The current study focuses on the detection and diagnosis of such difficult to detect faults while maintaining good detection accuracy for the other faults. The difficult to observe/detect faults will be referred to as incipient faults.

Lack of observability often arises due to the low signal to noise ratio in the measurements used for fault detection and diagnosis (FDD) and feedback control [16,17]. Specifically, the controller forces the controlled variables to remain close to their set-points at all times. Furthermore, with the addition of noise, the effects of faults are masked. In addition, the lack of distinguishability between different process faults is related to the fact that various process faults have a similar effect on the dynamic responses of the measured variables.

FDD algorithms that rely on data collected from the process operation are referred to as passive, while active FDD approaches have also been proposed to improve detection [18]. Active FDD involves injecting persistently exciting input signals into the system and using the resulting input–output data for incipient fault detection and diagnosis [19–21]. The disadvantage of active FDD is that it introduces an external disturbance to the process which may temporarily impact the operation, and thus, its use should be limited. To the knowledge of the authors, the combination of active and passive FDD approaches into one algorithm for detecting a mix of non-incipient and incipient faults have not been studied.

Following the above, the focus of the current work is on developing deep learning techniques for the detection of faults with an emphasis on the detection of incipient faults. However, faults and their effects on process variables are strongly coupled with each other. Thus, improving the detection of incipient faults should be achieved without degrading the detection of the regular faults. Toward this goal, a novel hierarchical classification strategy based on DNN models is proposed that involves identifying separate models for different subsets of faults with different degrees of difficulty to detect. A combination of both passive and active FDD approaches is used. The DNN models used for the passive FDD component are of Recursive Neural Network (RNN) type to exploit the dynamic information in the data. It is also demonstrated that the detection accuracy of most faults can be enhanced by increasing the time horizon of the LSTM-based model. While the passive approach is used in the higher level of the hierarchy, the active approach involving the injection of external signals is only used in the last level of the hierarchy for detecting incipient faults that cannot be diagnosed otherwise. It is shown that the passive FDD approach is effective for identifying most faults, but the active approach is required for detecting incipient faults.

All studies in this work are conducted with a standard set of simulated data from the Tennessee Eastman Process (TEP) for a fair comparison with several algorithms reported for this system [22–26,26]. Since its introduction, the TEP has served as a benchmark problem for testing control and fault detection algorithms, and it is thus ideal for comparing existing approaches to our proposed algorithm. It should be emphasized that due to the difficulty in detecting a set of incipient faults for TEP (faults 3, 9 and 15), many studies on FDD for this system were carried out by ignoring these faults altogether [23,24]. For those studies of FDD for the TEP process that consider all the faults together, the regular faults were detected with an acceptable level of success, but the detection of incipient faults was very inaccurate [27]. In this work, we address the gap related to the miss-classification of incipient faults by proposing a novel hierarchical structure that combines a deep learning approach with an active FDD approach. Furthermore, it was also demonstrated that just the hierarchical structure along with deep NNs are not enough for classifying incipient faults

through an ablation study. Thus, an active FDD (introduction of excitation signal) approach in combination with a hierarchical deep NN structure are both required for efficient fault diagnosis. Additional reported methods applied to TEP are further reviewed and discussed in Results and Discussions section (Sections 4 and 5). The comparison of our approach to several reported methods shows that our approach provides comparable or superior FDD accuracy for regular faults but clear superiority for incipient faults.

The main contributions of the current study are:

1.  A novel hierarchical structure was developed combining passive FDD with active FDD to enhance the detection and classification accuracy for incipient faults.
2.  Design of PRBS signal for improving the observability of the incipient faults.
3.  The LSTM model was optimized with respect to data horizon for better classification of faults.
4.  A comprehensive comparison of the proposed method to several other methods was carried out to demonstrate the efficacy of the proposed method for both regular and incipient faults.

The paper is organized as follows. Fundamentals used in the work are presented in Section 2. Explanation on the hierarchical structure of the proposed methodology is presented in Section 3. The results are presented in Section 4. Discussions and comparisons with previously reported approaches are presented in Section 5 followed by conclusions in Section 6.

## 2. Preliminaries

### 2.1. Recurrent Neural Networks (RNNs)

The current study uses a Recurrent Neural Network (RNN) type model that was originally developed for handling dynamic data by using time sequences of data $\mathbf{x}_t^i$, $t = 1, 2, \ldots, T \in \mathbb{R}^{d_h \times d_x}$ as inputs to the network [28]. Parameters associated with RNN are shared along a time horizon to capture temporal correlations in data. This enhances the generalization capability of the model to time sequences that were not used for model calibration. A well-known challenge for training RNNs is the vanishing gradient or exploding gradient problem arising from the use of gradient descent algorithms in combination with sigmoid activation functions [29]. To deal with this problem, the best practice is to use gated-type unit structures within RNN models such as Long-Short Term Memory units (LSTM) [30] and Gated Recurrent Units (GRU) [31]. LSTM is reviewed in the following sub-section since they serve as the basis for the models used in the current study for FDD.

### 2.2. Long Short-Term Memory (LSTM) Units

The LSTM unit is composed of three gated units and a memory cell [30]. Figure 1 shows a single LSTM unit that includes four major gates: the forget gate ($\mathbf{f}_t$), the input gate ($\mathbf{i}_t$), the output gate ($\mathbf{o}_t$) and the update gate ($\mathbf{g}_t$). The key component of the LSTM unit is the memory cell ($\mathbf{c}_t \in \mathbb{R}^{d_h \times 1}$) that is responsible for storing critical long-term dependencies learned over time. The input gate ($\mathbf{i}_t$) is responsible for evaluating which part, if any, of the past historical data should be kept. Thus, the function of the input gate is to allow the network to keep only relevant information from the previous time steps and discard the rest for a sample $i$.

Subsequently, the information that is worth recording is determined by the memory cell ($\mathbf{c}_t$). The process of identifying information and storing in the memory cell consists of two parts: new information that is recorded and information that is discarded. The information that should be discarded from previous cell state $\mathbf{c}_{t-1}^i$ is determined by the forget gate ($\mathbf{f}_t$), which is responsible for forgetting previously stored cell state values that have lost their relevance. Then, new relevant information is added, and existing cell-state values are updated by first selecting which values to update using the input gate $\mathbf{i}_t^i$, and the output from the input gate is then multiplied by the new information generated by the update gate $\mathbf{g}_t^i$. Ultimately, the output $\mathbf{h}_t$ is computed at every time step from the information contained in the memory cell and it is further gated by an output gate according to its importance or relevance. The mathematical equations describing these gating operations are as follows:

$$\mathbf{i}_t^i = \sigma(\mathbf{W}_i\mathbf{x}_t^i + \mathbf{R}_i\mathbf{h}_{t-1}^i + \mathbf{b}_i)$$
$$\mathbf{g}_t^i = \tanh(\mathbf{W}_g\mathbf{x}_t^i + \mathbf{R}_g\mathbf{h}_{t-1}^i + \mathbf{b}_g) \tag{1}$$

$$\mathbf{c}_t^i = \mathbf{f}_t^i \odot \mathbf{c}_{t-1}^i + \mathbf{i}_t^i \odot \mathbf{g}_t^i \tag{2}$$

where $\boldsymbol{\sigma}()$ and $\tanh()$ are the element-wise sigmoid and hyperbolic tangent functions, respectively.

$$\mathbf{o}_t^i = \sigma(\mathbf{W}_o\mathbf{x}_t^i + \mathbf{R}_o\mathbf{h}_{t-1}^i + \mathbf{b}_o)$$
$$\mathbf{h}_t^i = \mathbf{o}_t^i \odot \tanh(\mathbf{c}_t^i) \tag{3}$$

where $\mathbf{R} = [\mathbf{R}_f\,\mathbf{R}_i\,\mathbf{R}_g\,\mathbf{R}_o]^T \in \mathbb{R}^{4d_h \times d_h}$ are known as recurrent weights, $\mathbf{W} = [\mathbf{W}_f\,\mathbf{W}_i\,\mathbf{W}_g\,\mathbf{W}_o]^T \in \mathbb{R}^{4d_h \times d_x}$ are all the input weights, $\mathbf{b} = [\mathbf{b}_f\,\mathbf{b}_i\,\mathbf{b}_g\,\mathbf{b}_o]^T \in \mathbb{R}^{d_h \times 1}$ are the bias parameters.

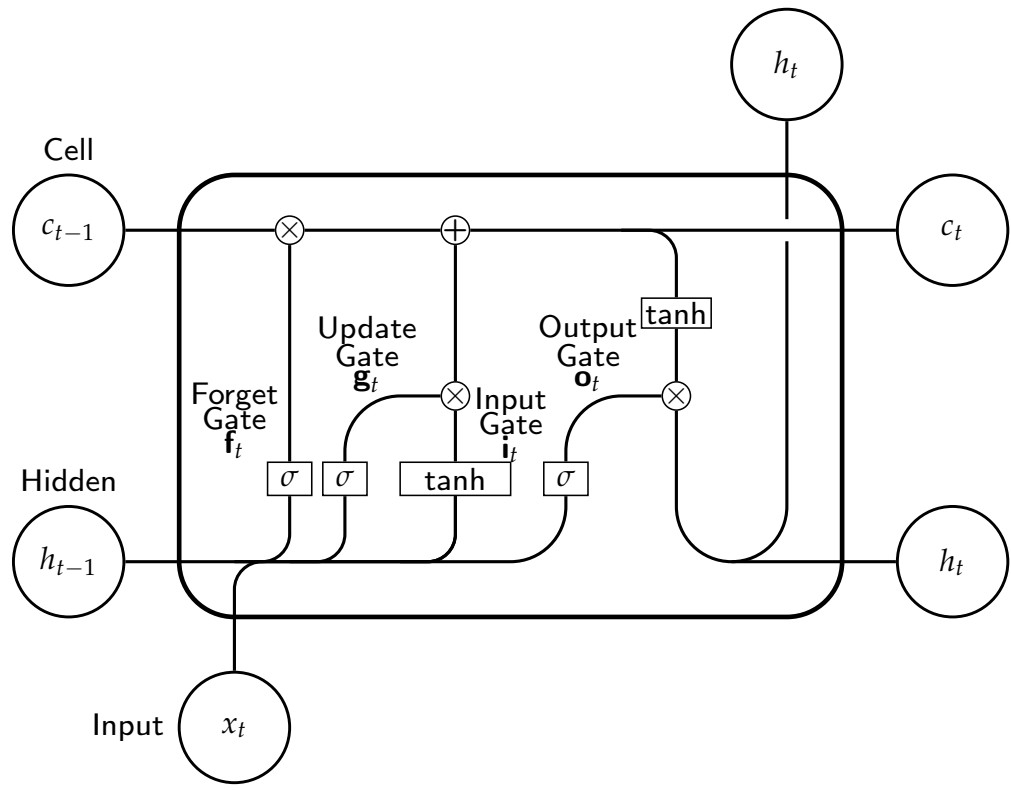

**Figure 1.** Schematic of an LSTM memory cell.

### 2.3. Deep LSTM Supervised Autoencoder Neural Network (DLSTM-SAE NN)

The training of a Deep Supervised Autoencoder Neural Network (DSAE-NN) model, as schematically shown in Figure 2, is based on the minimization of a weighted sum of the reconstruction loss function and the supervised classification loss corresponding to the first and second terms in Equation (4), respectively. Addition of the unsupervised loss function i.e., reconstruction loss function, improves the generalization of supervised autoencoder model [32]. Furthermore, it serves as the regularization term which constrains the problem in terms of latent variables, thus reducing over-fitting. Meanwhile, the minimization of the classification loss function, i.e., multi-class cross-entropy loss function, ensures the non-linear latent variables extracted are the predictors of the output label. The mean squared error function is used as a reconstruction loss and softmax cross-entropy is used as the classification loss. The overall goal is to learn a function that predicts the class labels in

one-hot encoded form $\mathbf{y}_i \in \mathbb{R}^m$ from inputs $\mathbf{x}_i \in \mathbb{R}^{d_x \times 1}$.

For training DSAE-NN, the following loss function is minimized:

$$l_{DSAE} = \frac{\lambda_1}{N}||\mathbf{x}_s - \hat{\mathbf{x}}_s||_2^2 + \frac{1}{N}\sum_{s=1}^{N}\sum_{c=1}^{m} -y_{s,c}log(p_{s,c}) \tag{4}$$

In this work, we use LSTM units instead of dense layers for both the encoder and decoder, as shown in Figure 3. The goal is to reconstruct and classify input sequences at time $t$ simultaneously. The encoder transforms the input time sequences using Equations (1)–(3) to learn important features and encode these features $\mathbf{z} \in \mathbb{R}^{d_h \times 1}$. The decoder function reconstructs the input using the extracted feature vectors. The operation performed by the encoder for a single LSTM layer between the input variables to the latent variables $\mathbf{z}_t^i \in \mathbb{R}^{d_h \times 1}$ can be mathematically described as follows:

$$\mathbf{z}_t^i = \zeta_e(\mathbf{x}_t^i) \tag{5}$$

The latent variables $\mathbf{z}_t^i$ are used both to predict the class labels and to reconstruct back the inputs $\mathbf{x}$ as follows:

$$\hat{\mathbf{x}_t^i} = \zeta_d(z_t^i) \tag{6}$$

$$\hat{\mathbf{y}_t^i} = f_c(\mathbf{W}_c\mathbf{z}_t^i + \mathbf{b}_c) \tag{7}$$

where $\zeta_e$ and $\zeta_d$ is the LSTM encoder and decoder function, respectively. $f_c$ is a non-linear activation function (softmax layer) for the output layer. $\mathbf{W}_c \in \mathbb{R}^{m \times d_z}$ and $\mathbf{b}_c \in \mathbb{R}^m$ are the output weight matrix and bias vector, respectively.

$$p_{s,c} = \frac{e^{(\hat{y}_{s,c})}}{\sum_{c=1}^{m} e^{(\hat{y}_{s,c})}} \tag{8}$$

where $\lambda_1$ is the weight multiplying the reconstruction loss $L_r$ in the cost to be minimized, $m$ is the number of classes, $y_{s,c}$ is a binary indicator (0 or 1) equal to 1 if the class label $c$ is the correct one for observation $s$ and 0 otherwise, $\hat{y_{s,c}}$ is the non-normalized log probabilities and $p_{s,c}$ is the predicted probability for a sample $s$ of class $c$. Moreover, to avoid overfitting, a regularization term is added to the objective function in Equation (4). Accordingly, the objective function for Deep LSTM SAE NNs used for FDD is as follows:

$$\min_{\mathbf{W}} l_{DLSTM-SAE} = \min \frac{1}{N}\left[\lambda_1||\mathbf{x}_s - \hat{\mathbf{x}}_s||_2^2 + \lambda_2\sum_{s=1}^{N}\sum_{c=1}^{m} -y_{s,c}log(p_{s,c}) + \lambda_3\sum_{L}\sum_{k}\sum_{j} \mathbf{W}_{kj}^{[L]2}\right] \tag{9}$$

where $\mathbf{W}_{kj}^{[L]}$ represents the weight matrices for each layer $L$ in the network and the weights on the individual objective functions $\lambda_1, \lambda_2, \lambda_3$ are chosen using validation data.

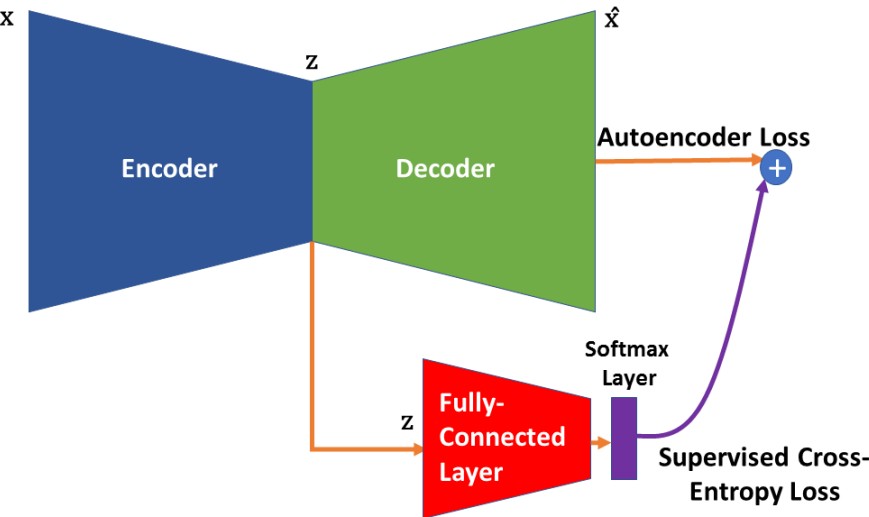

**Figure 2.** Schematic of a single layer Supervised Autoencoder Neural Network (SAE-NN).

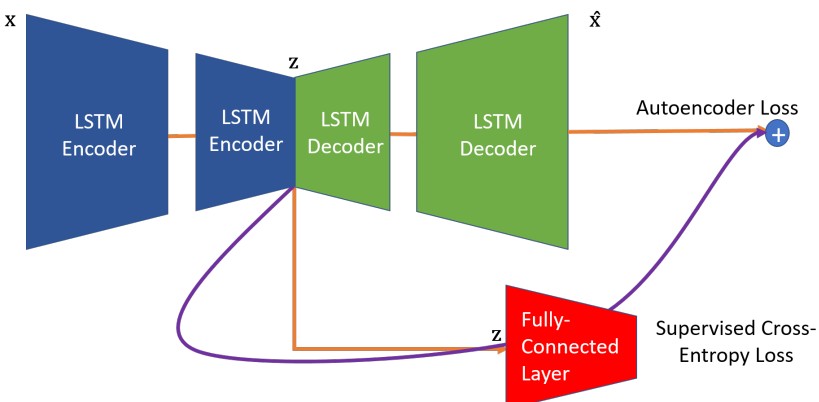

**Figure 3.** Schematic of a Deep LSTM Supervised Autoencoder Neural Network (DLSTM-SAE NN).

### 2.4. Model Structure and Specifications

The Deep LSTM-SAE model used in the current study was developed with training and testing data sets generated from the Tennessee Eastman Process (TEP: schematic shown in Figure 4) simulation. The data are extracted from simulations of the system conducted at either the normal state or when each of the 20 different faults is occurring in the process. It is assumed that at each sampling interval, 52 different variables (refer Table 1) are measured and organized into a vector. Each such vector of measurements is acquired every 3 min. It should be noticed that during testing of the methods proposed in this study, the normal state is considered as a different separate class, and hence, a total of 21 different classes (refer Table 2, i.e., 20 faulty plus one normal operations, are considered for classification. The standard dataset can be downloaded from http://depts.washington.edu/control/LARRY/TE/download.html (accessed on 9 April 2022). The simulator is ran for 72 h (training: 24 h; testing: 48 h) for each fault, generating 1440 samples for each fault class and normal class. The data are then divided between calibration and validation data sets where the first 480 samples are used as training data and the rest are used for testing for each class. This results in a total of 10,080 training samples and 19,200 testing samples. A small fraction of training dataset is used as validation dataset for selecting the optimal hyper-parameters. It is important to note that the number of training, validation and testing samples vary depending on the time horizon used in the DLSTM-SAE model. The results reported in the following section are based on the classification accuracy of the test dataset, i.e., on data that were not used for model calibration. The experiments in this paper have been implemented on an Intel Core i7-7700HQ PC (2.80 GHz, 16 GB RAM) and NVIDIA

GeForce GTX 1060 (6 GB) 64 Bit Windows 10 operating system in Python ® environment. The models are developed using Keras [33] (an open deep learning library) on TensorFlow platform [34]. All hyper-parameters such as the number of LSTM encoder layers, LSTM units in each layer, weights and learning rate are optimized using Keras tuner.

**Table 1.** Measured and manipulated variables (from Downs and Vogel, 1993).

| Variable Name | Variable Number | Units | Variable Name | Variable Number | Units |
|---|---|---|---|---|---|
| A feed (stream 1) | XMEAS (1) | kscmh | Reactor cooling water outlet temperature | XMEAS (21) | °C |
| D feed (stream 2) | XMEAS (2) | kg h$^{-1}$ | Separator cooling water outlet temperature | XMEAS (22) | °C |
| E feed (stream 3) | XMEAS (3) | kg h$^{-1}$ | Feed %A | XMEAS (23) | mol% |
| A and C feed (stream 4) | XMEAS (4) | kscmh | Feed %B | XMEAS (24) | mol% |
| Recycle flow (stream 8) | XMEAS (5) | kscmh | Feed %C | XMEAS (25) | mol% |
| Reactor feed rate (stream 6) | XMEAS (6) | kscmh | Feed %D | XMEAS (26) | mol% |
| Reactor pressure | XMEAS (7) | kPa guage | Feed %E | XMEAS (27) | mol% |
| Reactor level | XMEAS (8) | % | Feed %F | XMEAS (28) | mol% |
| Reactor temperature | XMEAS (9) | °C | Purge %A | XMEAS (29) | mol% |
| Purge rate (stream 9) | XMEAS (10) | kscmh | Purge %B | XMEAS (30) | mol% |
| Product separator temperature | XMEAS (11) | °C | Purge %C | XMEAS (31) | mol% |
| Product separator level | XMEAS (12) | % | Purge %D | XMEAS (32) | mol% |
| Product separator pressure | XMEAS (13) | kPa guage | Purge %E | XMEAS (33) | mol% |
| Product separator underflow (stream 10) | XMEAS (14) | m$^3$ h$^{-1}$ | Purge %F | XMEAS (34) | mol% |
| Stripper level | XMEAS (15) | % | Purge %G | XMEAS(35) | mol% |
| Stripper pressure | XMEAS (16) | kPa guage | Purge %H | XMEAS (36) | mol% |
| Stripper underflow (stream 11) | XMEAS (17) | m$^3$ h$^{-1}$ | Product %D | XMEAS (37) | mol% |
| Stripper temperature | XMEAS (18) | °C | Product %E | XMEAS (38) | mol% |
| Stripper steam flow | XMEAS (19) | kg h$^{-1}$ | Product %F | XMEAS (39) | mol% |
| Compressor Work | XMEAS (20) | kW | Product %G | XMEAS (40) | mol% |
| D feed flow | XMV (1) | kg h$^{-1}$ | Product %H | XMEAS (41) | mol% |
| E feed flow | XMV (2) | kg h$^{-1}$ | A feed flow | XMV (3) | kscmh |
| A + C feed flow | XMV (4) | kscmh | Compressor recycle valve | XMV (5) | % |
| Purge valve | XMV (6) | % | Separator pot liquid flow | XMV (7) | m$^3$h$^{-1}$ |
| Stripper liquid product flow | XMV (8) | m$^3$h$^{-1}$ | Stripper steam valve | XMV (9) | % |
| Reactor cooling water flow | XMV (10) | m$^3$h$^{-1}$ | Condenser cooling water flow | XMV (11) | m$^3$h$^{-1}$ |

**Table 2.** Process faults for classification in the TE process.

| Fault | Description | Type |
|---|---|---|
| IDV(1) | A/C feed ratio, B composition constant (stream 4) | step |
| IDV(2) | B composition, A/C ratio constant (stream 4) | step |
| IDV(3) | D Feed temperature | step |
| IDV(4) | Reactor cooling water inlet temperature | step |
| IDV(5) | Condenser cooling water inlet temperature (stream 2) | step |
| IDV(6) | A feed loss (stream 1) | step |
| IDV(7) | C header pressure loss reduced availability (stream 4) | step |
| IDV(8) | A, B, C feed composition (stream 4) | random variation |
| IDV(9) | D feed temperature | random variation |
| IDV(10) | C feed temperature (stream 4) | random variation |
| IDV(11) | Reactor cooling water inlet temperature | random variation |
| IDV(12) | Condenser cooling water inlet temperature | random variation |
| IDV(13) | Reaction kinetics | slow drift |
| IDV(14) | Reactor cooling water | valve sticking |
| IDV(15) | Condenser cooling water valve | stiction |
| IDV(16) | Deviations of heat transfer within stripper | random variation |
| IDV(17) | Deviations of heat transfer within reactor | random variation |
| IDV(18) | Deviations of heat transfer within condenser | random variation |
| IDV(19) | Recycle valve of compressor, underflow stripper and steam valve stripper | stiction |
| IDV(20) | unknown | random variation |

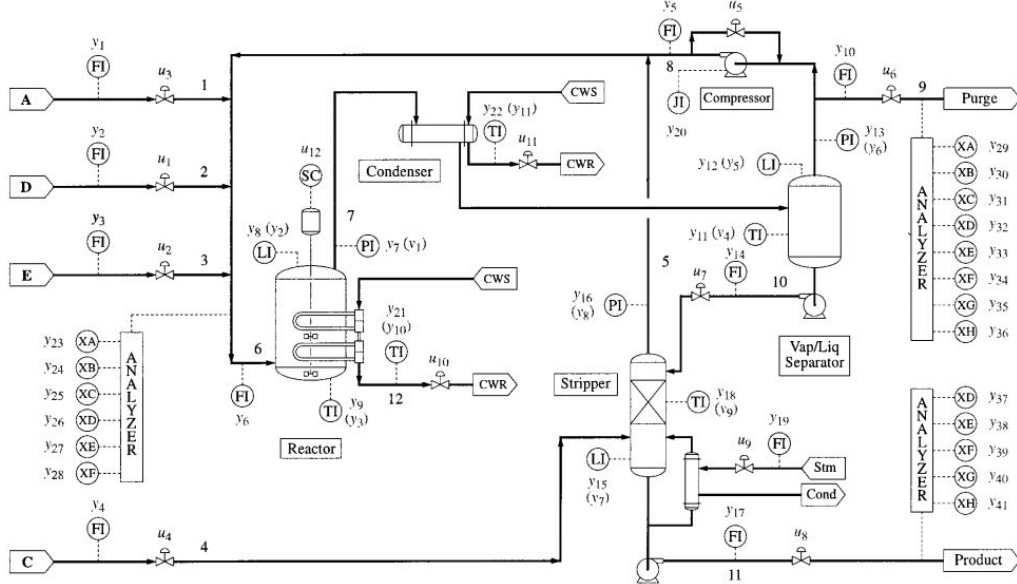

**Figure 4.** Schematic of Tennessee Eastman Process.

## 3. Hierarchical Structure

The key goal of the work is to improve the detection and diagnosis of incipient faults but without sacrificing the detection accuracy for the regular (non-incipient) faults. Thus, we need to increase the sensitivity of the non-linear FDD algorithm with respect to the incipient faults but without losing sensitivity with respect to the non-incipient faults. The sensitivity of non-linear models such as deep neural networks is highly dependent on the variability of the data used for calibration. Accordingly, a key data pre-processing step toward model calibration involves data standardization, i.e., mean centering and normalization. It is hypothesized that by building separate models for different groups of faults, it is possible to increase the sensitivity of different models and distinguishability between faults because of the different re-normalization conducted within each group.

Following the above, a hierarchical structure is proposed as shown in Figure 5. This structure includes the following sequential steps for training of the model with a training data set:

1. The training data are mean centered and normalized.
2. The faults are classified into two groups: group 1—easily distinguishable faults and group 2—difficult to distinguish faults, which include the incipient faults along with normal operation data class.
3. A Deep LSTM-SAE model denoted as M1 is designed for identifying the faults of group 1 or identifying all faults in group 2 as a single fault.
4. The data for group 2 identified in the previous step are mean centered and re-normalized.
5. A neural network model is designed specifically for group 2 denoted as M2.
6. For faults that are not accurately identified by M2, a PRBS is designed and injected into locations in the system that are informative about these faults.

Based on the trained hierarchical structure, online detection and diagnosis for any new sample proceeds as follows:

1. The data corresponding to the sample is mean centered and normalized as in step 1 of the training procedure.
2. The sample is classified as either in group 1 of easy to observe faults or group 2 of difficult to identify faults.
3. If sample is in group 1, it is classified accordingly by model M1. If it is in group 2, it is re-normalized according to the re-normalization in step 4 of the training procedure.

4.  If the sample is within group 2, it is identified by model M2 in step 5 of the training procedure.
5.  If the sample is not identified accurately by the model for group 2, PRBS signals are injected as specified in step 6 of the training procedure, and the corresponding faults are diagnosed from the resulting data.

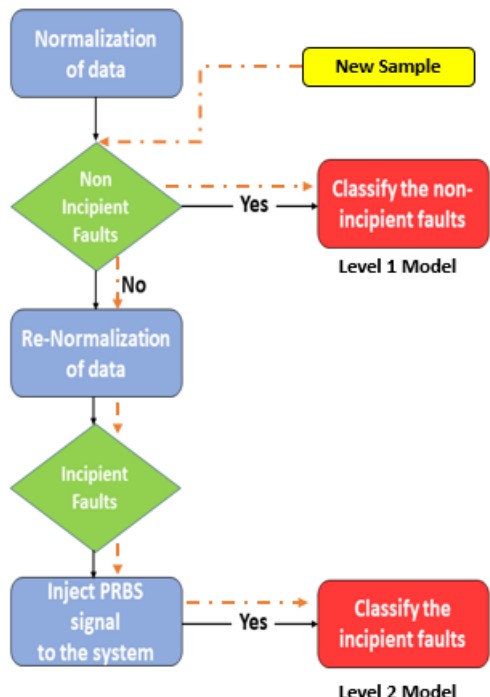

**Figure 5.** Hierarchical structure used for fault detection and diagnosis.

It should be noticed that in this algorithm, the normal operation is treated as an additional fault class denoted as Class 1. The incipient faults are characterized by responses that are very similar to the normal state (TEP: faults 3, 9 and 15). It should also be noted that the incipient faults are grouped along with the normal state as per step 1 of the training procedure; it may also result in miss-classification as other faults. Hence, the overall classification accuracy for the incipient faults must be assessed after the execution of the entire hierarchical procedure.

For model M1, the normalized data are fed to a first-level model where the softmax layer of LSTM-SAE NN uses 18 units instead of the 21 units (incipient faults and normal state grouped as one) as used in the non-hierarchical type model. The structure of model M2 is similar to model M1, but the difference is that the softmax layer involves only 4 units each for one of the incipient faults (3, 9, 15) and for the normal state (fault 0). The PRBS is injected only when the incipient fault cannot be properly identified with either models M1 or M2. Additional details about the PRBS signal design are given in the following section.

*Design: Pseudo-Random Binary Signal (PRBS)*

Although the hierarchical structure proposed in the previous section enhances the diagnosibility of few faults, the detection of incipient faults is still challenging due to the lack of excitation to detect these faults in the presence of noise. This problem is particularly acute in the TEP since the dataset contains variables that are used in closed-loop control, thus exhibiting a small variation with respect to their set-point values, making it difficult to estimate the occurrence of faults from such variables. To increase the diagnosibility of incipient faults, the use of active fault detection, as reviewed in the Introduction, is proposed for the TEP process. The lack of diagnosibility/distinguishability of the incipient faults

can be viewed as a problem of inaccurate identification of a model relating variability in measured values to faults. To improve the identification accuracy, it is required to use inputs that sufficiently excite the system dynamics in the presence of noise [35], which will result in larger changes in the measured quantities and larger sensitivity to fault changes. Thus, it is required to introduce additional excitation to the one available in regular operation of the system. Accordingly, external forcing signals are injected at particular points of the control loops, e.g., an excitation signal to the set-points of the loops that involve variables related to the difficult to detect faults. The addition of such excitation signals in combination with a separate deep neural network model (second level) in the hierarchical structure described in the previous section is investigated in the current study for detecting and diagnosing incipient faults that cannot be accurately identified with the regular operating data collected from the process.

To avoid a large negative impact of the external signals on the profitability of the plant, the input signals should meet certain constraints as follows:

1. Reduce input move sizes (to reduce wear and tear on actuators).
2. Reduce input and output amplitudes, power, or variance.
3. Short experimental time to prevent losses

In a practical implementation, the added excitation signal should result in variations in the measured quantities that will be large in magnitude relative to the noise. Toward this goal, it is necessary to include information of frequencies lower than the crossover frequency of the closed loop transfer function [36]. PRBS signals are used as excitation signals in this study, since they have a finite length that can be synthesized repeatedly with simple generators while presenting favorable spectra. The spectrum at low frequencies are flat and constant, while at high frequencies, the spectra drop off. Thus, the PRBS can be designed to have a specific bandwidth, which can be utilized for exciting the processes within the required range of frequencies [37]. The analytical expression for the power spectrum of a PRBS is given by:

$$s(\omega) = \frac{A^2(R+1)t_{cl}}{R}\left[\frac{\sin \omega t_{cl}/2}{\omega t_{cl}}\right]^2 \tag{10}$$

where $\omega$ is the frequency, $t_{cl}$ is the clock period (minimum time between a change in levels) which is a multiple of the sampling time ($T_s$) and $A$ is the amplitude of the signal. The sequence repeats itself after $T = R \times t_{cl}$ units of time, where $R = 2n - 1$ and $n$ is the number of shift registers used to generate the sequence. Thus, for designing the PRBS signal, it is necessary to estimate the amplitude and the frequency range.

$$\frac{2\pi}{T} \leq \omega \leq \frac{2.8}{t_{cl}} \tag{11}$$

Rivera and Gaikwad (1995) [36] Lee and Rivera, 2005 [38] and Garcia-Gabin and Lundh [37] provided practical guidelines for estimating the range of frequency needed for process closed-loop identification using time-domain information. The primary frequency band of interest for excitation is determined by the dominant time constants of the system.

$$\omega_{low} = \frac{1}{S_f \, t^{ol}} \tag{12}$$

where $t^{ol} = 4\tau^{ol} + t_d^{ol}$

$$\omega_{high} = \frac{4S_f}{t^{cl}} \tag{13}$$

$$\omega_{high} \leq \omega_N \tag{14}$$

where $S_f$ is a safety factor used to augment the bandwidth of the excitation signal, $t^{ol}$ is the open loop settling time and $t^{cl}$ is the settling time of the closed loop process without considering the time delays. $t^{ol}_d$ is the time delay of the open loop process. In addition, the upper value of the frequency must be lower than the Nyquist frequency $\omega_N$ to avoid aliasing. Although the magnitude of the signal has not been optimized in the current work, it could be further optimized by taking a profit function of the plant into consideration for minimal losses and using the validation data used for the FDD model.

## 4. Results

In this section, the industrial benchmark TEP is used to validate and demonstrate the effectiveness of the proposed method. Three tables are presented in this section to summarize the results. The tables show comparisons based on a standard set of simulated data from the TEP between our proposed algorithm with several algorithms reported in the literature. First, fault detection rates for all non-incipient faults are shown in Table 3 for different linear multivariate methods and DL-based methodologies. Secondly, Table 4 shows comparisons with those results that consider incipient fault detection for TEP along with non-incipient faults, i.e., all faults.

**Table 3.** Comparison of Fault Detection Rate with different methods with non-incipient faults only.

| Fault | PCA (15 comp.) | | DPCA (22 comp.) | ICA (9 comp.) | | DL (2017) | DL (2017) | DL (2018) | DL (2018) | DL (2019) | Proposed DL |
|---|---|---|---|---|---|---|---|---|---|---|---|
| | $T^2$ | SPE | $T^2$ | $I^2$ | AO | SAE-NN | DSN | GAN | OCSVM | CNN | Deep LSTM-SAE |
| 1 | 99.2% | 99.8% | 99% | 100% | 100% | 77.6% | 90.8% | 99.62% | 99.5% | 91.39% | 100% |
| 2 | 98% | 98.6% | 98% | 98% | 98% | 85% | 89.6% | 98.5% | 98.5% | 87.96% | 100% |
| 4 | 4.4% | 96.2% | 26% | 61% | 84% | 56.6% | 47.6% | 56.25% | 50.37% | 99.73% | 100% |
| 5 | 22.5% | 25.4% | 36% | 100% | 100% | 76% | 31.6% | 32.37% | 30.5% | 90.35% | 100% |
| 6 | 98.9% | 100% | 100% | 100% | 100% | 82.8% | 91.6% | 100% | 100% | 91.5% | 100% |
| 7 | 91.5% | 100% | 100% | 99% | 100% | 80.6% | 91% | 99.99% | 99.62% | 91.55% | 100% |
| 8 | 96.6% | 97.6% | 98% | 97% | 97% | 83% | 90.2% | 97.87% | 97.37% | 82.95% | 100% |
| 10 | 33.4% | 34.1% | 55% | 78% | 82% | 75.3% | 63.2% | 50.87% | 53.25% | 70.05% | 42.8% |
| 11 | 20.6% | 64.4% | 48% | 52% | 70 | 75.9% | 54.2% | 58% | 54.75% | 60.16% | 100% |
| 12 | 97.1% | 97.5% | 99% | 99% | 100% | 83.3% | 87.8% | 98.75% | 98.63% | 85.56% | 100% |
| 13 | 94% | 95.5% | 94% | 94% | 95% | 83.3% | 85.5% | 95% | 94.87% | 46.92% | 100% |
| 14 | 84.2% | 100% | 100% | 100% | 100% | 77.8% | 89% | 100% | 100 % | 88.88% | 100% |
| 16 | 16.6% | 24.5% | 49% | 71% | 78% | 78.3% | 74.8% | 34.37% | 36.37% | 66.84% | 100% |
| 17 | 74.1% | 89.2% | 82% | 89% | 94% | 78% | 83.3% | 91.12% | 87.25% | 77.11% | 100% |
| 18 | 88.7% | 89.9% | 90% | 90% | 90% | 83.3% | 82.4% | 90.37% | 90.12% | 82.74% | 100% |
| 19 | 0.4% | 12.7% | 3% | 69% | 80% | 67.7% | 52.4% | 11.8% | 3.75% | 70.87% | 40.4% |
| 20 | 29.9% | 45% | 53% | 87% | 91% | 77.1% | 44.1% | 58.37% | 52.75% | 72.88% | 100% |
| Average | 61.77% | 74.72% | 72.35% | 87.29% | 91.70% | 77.7% | 76.84% | 74.04% | 62.78% | 85.47% | 93.13% |

**Table 4.** Comparison of Fault Detection Rate with different methods (with all faults).

| Fault | DL (2017) | DL (2017) | DL (2018) | DL (2018) | DL (2019) | DL (2018) | DL (2021) | Proposed DL |
|---|---|---|---|---|---|---|---|---|
| | SAE-NN | DSN | GAN | OCSVM | CNN | Optimized LSTM | LSTM (attention) | Deep LSTM-SAE |
| 1 | 77.6% | 90.8% | 99.62% | 99.5% | 91.39% | 68% | 100% | 100% |
| 2 | 85% | 89.6% | 98.5% | 98.5% | 87.96% | 78% | 89% | 100% |
| 3 | 79.4% | 14.4% | 10.375% | 7.62% | 50.59% | 45% | 94% | 81.58% |
| 4 | 56.6% | 47.6% | 56.25% | 50.37% | 99.73% | 75% | 99% | 100% |
| 5 | 76% | 31.6% | 32.37% | 30.5% | 90.35% | 45% | 94% | 100% |
| 6 | 82.8% | 91.6% | 100% | 100% | 91.5% | 75% | 100% | 100% |
| 7 | 80.6% | 91% | 99.99% | 99.62% | 91.55% | 89% | 100% | 100% |
| 8 | 83% | 90.2% | 97.87% | 97.37% | 82.95% | 100% | 99% | 100% |
| 9 | 50.6% | 16.3% | 8.625% | 7.125% | 49.53% | 89% | 81% | 99.38% |
| 10 | 75.3% | 63.2% | 50.87% | 53.25% | 70.05% | 71% | 99% | 42.84% |
| 11 | 75.9% | 54.2% | 58% | 54.75% | 60.16% | 67% | 88% | 100% |
| 12 | 83.3% | 87.8% | 98.75% | 98.63% | 85.56% | 77% | 99% | 100% |
| 13 | 83.3% | 85.5% | 95% | 94.87% | 46.92% | 83% | 89% | 100% |
| 14 | 77.8% | 89% | 100% | 100 % | 88.88% | 56% | 99% | 100% |
| 15 | 55.5% | 26.7% | 12.5% | 14% | 43.54% | 89% | 22% | 100% |
| 16 | 78.3% | 74.8% | 34.37% | 36.37% | 66.84% | 99% | 31% | 100% |
| 17 | 78% | 83.3% | 91.12% | 87.25% | 77.11% | 0% | 97% | 100% |
| 18 | 83.3% | 82.4% | 90.37% | 90.12% | 82.74% | 89% | 95% | 100% |
| 19 | 67.7% | 52.4% | 11.8% | 3.75% | 70.87% | 20% | 97% | 40.4% |
| 20 | 77.1% | 44.1% | 58.37% | 52.75% | 72.88% | 88% | 85% | 100% |
| Average | 75.355% | 65.32% | 64.51% | 62.78% | 79.84% | 70.15% | 87.85% | 93.23% |

Finally, a systematic ablation study is conducted in Table 5 to demonstrate gradual improvements following the proposed methodology. Thus, in this table, the different levels of the proposed hierarchical algorithm are added one by one to observe their relative contribution to the FDD accuracy. In-depth details about the comparisons and ablation study are discussed in the next section.

**Table 5.** Ablation study for the proposed method.

| Faults | Non-Hierarchical DL NN | Hierarchical DL NN (No PRBS) | Hierarchical DL NN+ PRBS Addition for Fault 15 | Hierarchical + PRBS Addition for Fault 15 and Fault 9 |
|---|---|---|---|---|
| Fault 3 | 36% | 42% | 88.7% | 81.5% |
| Fault 9 | 32% | 18% | 38.4% | 99.3% |
| Fault 15 | 12% | 30% | 99.4% | 100% |
| Normal Operation | 18% | 25% | 100% | 98.1% |
| Average of all other Faults | 85% | 87% | 93.1% | 93.1% |
| Averaged Test Accuracy | 73.4% | 75.90% | 90.9% | 93.4% |

## 5. Discussions

We investigated the multi-class classification performance using a total of 20 fault modes presented in Table 2 which involve all of the compositions, manipulated and measurement variables in the TE process (Table 1). For an individual class IDV(i), the performance was typically evaluated by a confusion matrix which consists of true positives ($TP_i$), false positives ($FP_i$), true negatives ($TN_i$) and false negatives ($FN_i$). The notation used in the confusion matrix (refer Table 6) is as follows:

**Table 6.** Confusion matrix for each fault (IDV(i)).

| | Counts of Predicted Label i | Counts of Predicted Label other than i |
|---|---|---|
| Counts of real label *i* | $TP_i$ | $TN_i$ |
| Counts of real label other than *i* | $FP_i$ | $FN_i$ |

Two main important metrics for quantifying the performance of the proposed process monitoring methodology are as follows:

- Fault Detection Rate (FDR):

$$\text{FDR} = \frac{\text{number of fault data that have been detected as fault}}{\text{total number of faulty samples}}$$
$$= \frac{TP_i}{TP_i + FP_i} \tag{15}$$

FDR represents the probability that the abnormal conditions are correctly detected, which is an important criterion to compare between different methods in terms of their detection efficiency. Evidently, a very high FDR is desirable.

- False Alarm Rate (FAR):

$$\text{FAR} = \frac{\text{number of normal data that have been detected as fault}}{\text{total number of normal samples}}$$
$$= \frac{FP_i}{TP_i + TN_i} \tag{16}$$

where the class corresponding to normal operation is considered as the positive class. FAR represents the probability that the normal operation is wrongly identified as abnormal, and thus, a very low FAR is desired and necessary.

The fault detection results obtained with the hierarchical LSTM SAE NN model are compared with both linear multivariate statistical methods and deep learning methods reported in previous studies. For a fair comparison between the methods, for studies where only non-incipient faults were considered, the results were compared to fault detection results obtained from the first level of the hierarchical structure model, whereas for studies where all the faults were considered, the comparisons were made for results obtained from the second level of the hierarchical structure model. The Fault Detection Rate (FDR) for all the faults is compared for the proposed method, PCA [23], DPCA [23], ICA [24], Convolutional NN (CNN) [25], Deep Stacked Network (DSN) [26], Stacked Autoencoder (SAE) [26], Generative Adversarial Network (GAN) [22] and One-Class SVM (OCSVM) [22]. The Fault Detection Rates for all non-incipient faults and incipient faults are shown in Tables 3 and 4, respectively, for different methodologies along with the results from the proposed method. It can been seen from Table 3 that the proposed method outperformed the linear multivariate methods and other DL-based methods for most fault modes. For example, for PCA with 15 principal components, the average fault detection rates are 61.77% and 74.72% using the $T^2$ and $Q$ statistic, respectively. Since the principal components extracted using PCA capture static correlations between variables, DPCA is used to account for temporal correlations (both auto-correlations and cross-correlations) in the data. The effect of increasing the number of time samples in the Tennessee Eastman simulation is also investigated following the hypothesis that increasing the time horizon will enhance classification accuracy. In the case of DPCA, the number of lags used in the observation matrix is a key parameter. Since DPCA is only a data compression technique, it must be combined with a classification model for the purpose of fault detection. Accordingly, the output features from the DPCA model are fed into an SVM model that is used for final classification. Different time horizons were tried for training the DPCA model. Based on validation results, the best DPCA model was obtained with 22 lags. The average detection rate obtained was 72.35%. The ICA [24]-based monitoring scheme performs better than both PCA and DPCA-based methods with an averaged accuracy of approximately 90%. It should be noted that all these methods (PCA, DPCA and ICA) perform poorly for detecting incipient faults.

In addition to the comparison to linear methods, the proposed methodology was also compared with different DNN architectures such as CNN [26], DSN [26], SAE-NN (results reported in Chadha and Schwung, 2017 [26]), GAN [22], and OCSVM (results reported in Spyridon and Boutalis, 2018 [22]) reported previously. It can be seen that the proposed method also outperforms these DNN-based methodologies. The relative advantage of our method versus these other DNN architectures (Table 4) is due to the inclusion of the incipient faults within the normal class (hierarchical structure) and the supervised autoencoder (SAE) DNN architecture. This reduces the confusion between the normal samples with other non-incipient faults. However, the additional advantage of the proposed method over the other DNN architectures is realized when the hierarchical structure is used in combination with the PRBS signals as further discussed below. It should be noted that all these comparisons were based on an identical data set. Similarly, the fault detection rate for all faults is compared with different DL-based models in Table 4 including SAE-NN, DSN, GAN, OCSVM, CNN, Optimized LSTM [39] and LSTM along with an attention mechanism [27]. It can be seen that the proposed methodology improves the averaged test classification accuracy for all faults significantly.

Subsequently, the faults were diagnosed using the proposed hierarchical structure where the first level model of the hierarchical structure classifies non-incipient faults and the second level model classifies incipient faults. For the first level model, there are 7382 training samples and 17,442 testing samples in total with a time horizon of 150 timesteps. The model consists of 182 encoder LSTM units, which is followed by 116 LSTM units

for processing of the output of the encoding layer. Thereafter, the output of the second LSTM layer is passed through a dense layer for classification. Hyper-parameters such as number of layers, number of LSTM units in each layer, classification weights, learning rate, time-horizon etc. are selected using validation data that are part of the training dataset. The confusion matrix for level 1 model is presented in Figure 6. The hyper-parameter search is implemented using a Keras-tuner. Firstly, a grid of hyper-parameters is defined, for example the number of encoder layers = [1, 2, 3], number of LSTM units for each of these layers ranging from 2 to 200 with an interval of 2 = [10:2:200], learning rate = [0.1, 0.2, 0.3, 0.01], value of weights in the objective function, etc. Keras-tuner trains the model using different combinations of these hyper-parameters values, and the averaged validation accuracy is evaluated at every epoch. The models are trained with a few epochs at the start, and the selected models with high validation accuracy are chosen to be trained for more epochs. A study was conducted to select the optimal time horizon for the LSTM-based model. It can be seen from Figure 7 that the classification averages can be enhanced by extending the length of the time horizon of past data fed to the LSTM-based model. A total of 150 time steps were chosen as the optimal time-horizon.

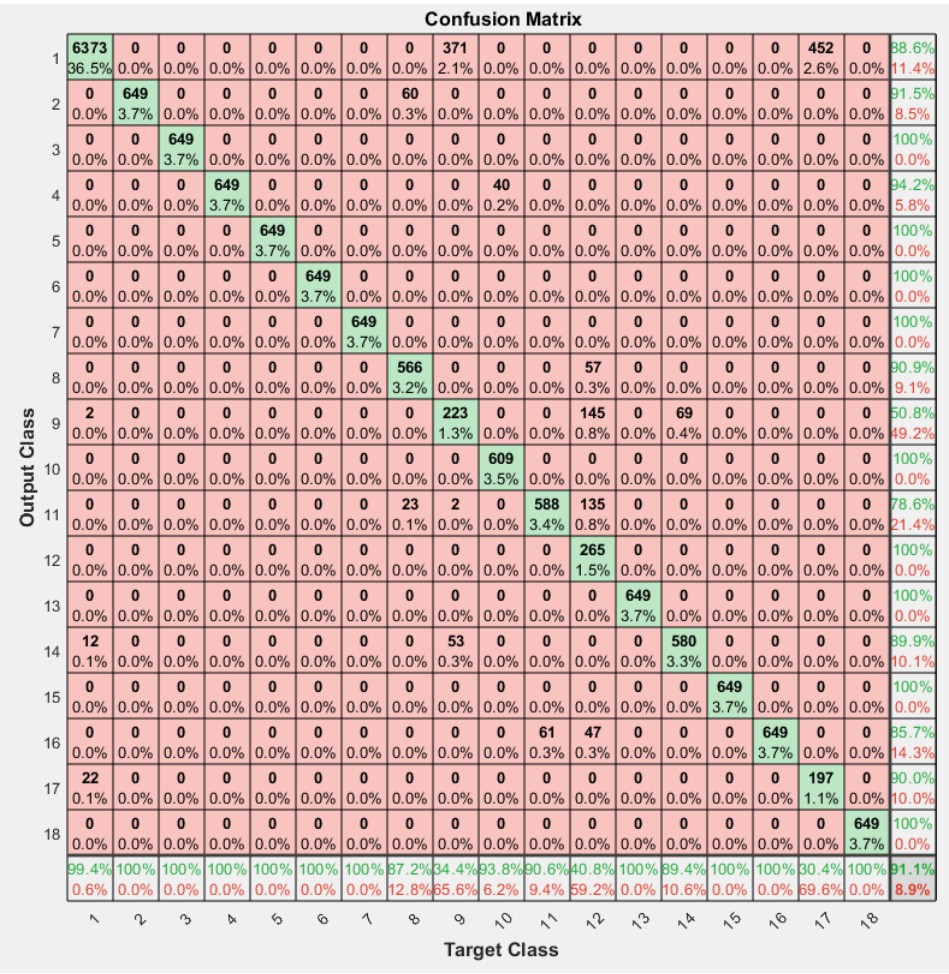

**Figure 6.** Confusion matrix for the first level model of the hierarchical structure (i.e., classification of non-incipient faults and considering incipient faults as a normal class).

The next important design parameter for the second level hierarchical model is the location in the process at which the external excitation signal should be introduced to maximize information about the occurring incipient fault. In this work, this choice is based on the flow-sheet and by identifying which variables are mostly correlated to the incipient faults under consideration. Specifically, the excitation signals were added to

process set-points in control loops that are most correlated to the incipient faults. When the selection of the variable to be excited by a PRBS is not obvious from the process flow-sheet, a more systematic approach is to use sensitivity analysis, e.g., the sensitivity of changes in the variable connected to the fault to all process variables. Since it may be detrimental to perturb the set-point continuously by the PRBS signal, the latter can be introduced intermittently into the process. In the current work, an excitation signal of length 40 time-steps was intermittently introduced every 4 h into the process by assuming that such an event will not impact significantly the profitability of the process (for test data). Changes in the separator temperature set-point will force changes in the condenser temperature. Since the fault to be identified is stiction in the valve that affects the condenser temperature, the imposed PRBS in the separator set-point indirectly helps in identifying fault 15. For fault 9, i.e., a random variation in D feed temperature (refer to Table 2), the PRBS excitation ($\omega \in [\omega_{cl}, \omega_n]$ where $\omega_{cl} = 0.0087$ rad/s and $\omega_n = 1.74$ rad/s) signal is introduced to the D feed ratio in order to create a suitable excitation. After developing this PRBS signal, we added both signals to the process at different times during the simulation. For fault 15, the PRBS signal is designed with a frequency range of $\omega \in [\omega_{cl}, \omega_n]$ where $\omega_{cl} = 0.005$ rad/s and $\omega_n = 1.74$ rad/s.

A systematic ablation study is conducted in Table 5 in order to demonstrate the gradual improvements in the results by showing fault detection rates of incipient faults, normal operation and non-incipient faults for 4 cases: i—without the hierarchical structure with one DL model, ii—with the hierarchical structure and iii—with the hierarchical structure and with the addition of one PRBS signal related to fault 15 and iv—with the hierarchical structure and with the addition of two PRBS signals related to fault 9 and fault 15. Other than a slight decrease in the detection of the Normal operation with the hierarchical structure and the addition of the two PRBS signals, the improvements in all other faults and in the average test accuracy are evident.

For the second level model, there are 1796 training samples and 4196 testing samples in total with a time horizon of 150 time-steps. The model consists of 284 encoder LSTM units in the first hidden layer, and the second layer consists of 100 LSTM units, which is followed by 278 LSTM units for processing of the output of the encoding layer. Thereafter, the output of the third LSTM layer is passed through a dense layer for classification. Hyper-parameters such as the number of layers, number of LSTM units in each layer, classification weights, learning rate, time-horizon, weights in the loss function, etc. are selected using the validation data which is part of the training dataset. The hyper-parameter search is implemented again using the Keras-tuner. For the second level model, the samples corresponding to fault 0 (normal) and incipient faults are considered. Figure 8 shows the confusion matrix after introducing the PRBS signal that was designed for identifying fault 15 and Figure 9 shows the confusion matrix after introducing both PRBS signals that were designed for identifying fault 15 and fault 9. The total FAR calculated using Equation (16) was 2.41%.

The averaged fault classification rates for all non-incipient faults and for all faults (including incipient faults) are shown in Figures 10 and 11, respectively. Figure 10 shows a bar-chart comparison of the proposed method with several non-linear methods such as sparse representation [40], SVM [41], hierarchical model based method [42], Random Forest, and structural SVM. It can be seen that the hierarchical deep RNN-based method outperforms other methods with a significant margin. It should be noted that the comparisons made in Figure 10 do not consider incipient faults. In Figure 11, the averaged test accuracy of all faults (both incipient and non-incipient faults) are compared with other DL-based methods [43]. It can be seen that the second level hierarchical model combined with the introduction of the designed PRBS signals significantly improves the classification of the incipient faults, and thus, the averaged test accuracy for fault diagnosis increases significantly.

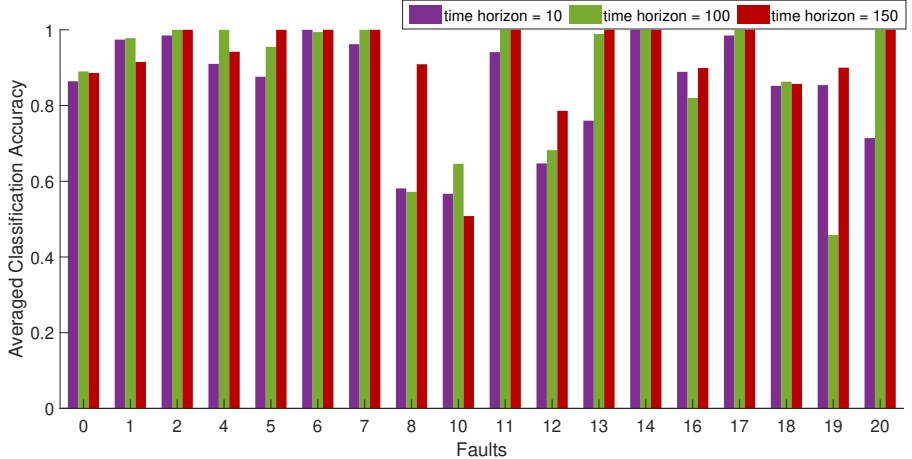

**Figure 7.** Selection of optimal time horizon for Hierarchical LSTM-SAE Level 1 model.

**Confusion Matrix**

True Label

|      |       |       |   |
|------|-------|-------|---|
| 1    | 0     | 0     | 0 |
| 0.06 | 0.887 | 0.051 | 0 |
| 0.34 | 0.271 | 0.384 | 0 |
| 0    | 0     | 0     | 1 |

Predicted Label

**Figure 8.** Confusion matrix on test data for the second level model of the hierarchical structure: after adding designed PRBS signal with respect to fault 15.

**Confusion Matrix**

True Label

|      |       |       |   |
|------|-------|-------|---|
| 0.98 | 0.018 | 0     | 0 |
| 0.18 | 0.815 | 0     | 0 |
| 0    | 0.006 | 0.994 | 0 |
| 0    | 0     | 0     | 1 |

Predicted Label

**Figure 9.** Confusion matrix on test data for the second level model of the hierarchical structure: after adding designed PRBS signal with respect to fault 9 and fault 15.

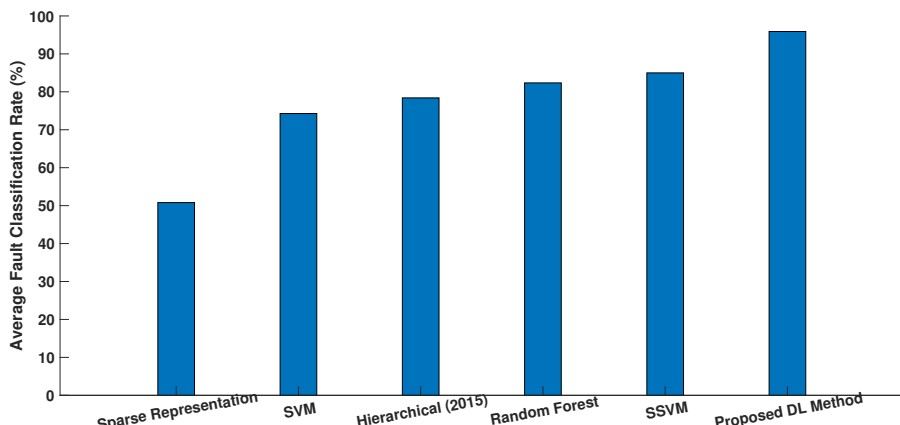

**Figure 10.** Comparison of averaged fault classification rates (non-incipient faults only).

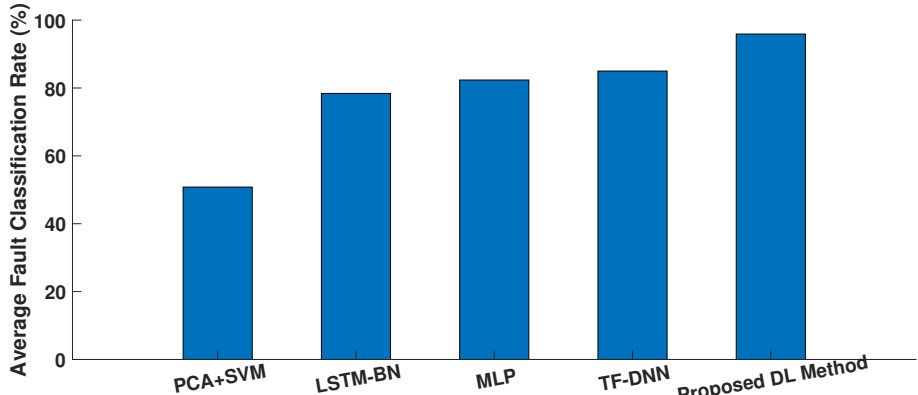

**Figure 11.** Comparison of averaged fault classification rates (all faults).

### 6. Conclusions

This work studied the application of a deep learning model within a hierarchical structure as a way to increase the detection and classification of faults in the Tennessee Eastman Process (TEP). The TEP simulation contains 20 different faults that were used during this study to make the classification problem. As previously reported by other researchers, a subset of these faults—referred to in this study as incipient—is particularly difficult to diagnose due to low signal-to-noise ratio and similarities in the resulting dynamic responses corresponding to different faults.

A comparison between deep learning techniques to a multivariate linear technique for fault detection such as PCA, DPCA, ICA and other deep learning methods is also presented. It is observed that a hierarchical LSTM-based model is superior to traditional linear and other deep learning-based methods for fault classification due to their ability to capture nonlinear dynamic behavior. It was also shown that the classification averages can be enhanced by extending the length of the time horizon of past data fed to the RNN-based model. However, most of these improvements in classification occurred for the non-incipient faults. Therefore, an active fault detection approach was pursued where a hierarchical model structure combined with external PRBS signals was proposed that proved to be particularly effective for classifying incipient faults. Future studies will address the trade-off between the impact of the injected PRBS signals on quality and productivity versus the benefit from the early detection of incipient faults.

**Author Contributions:** P.A.: methodology, formal analysis, data curation, writing—review and editing, software, visulaization; J.I.M.G.: methodology, writing and editing; A.E.: writing—review, supervision; H.B.: methodology, writing-editing and review, supervision, project administration, funding acquisition. All authors have read and agreed to the published version of the manuscript.

**Funding:** This work is the result of the research project supported by MITACS grant IT10393 through MITACS-Accelerate Program.

**Institutional Review Board Statement:** Not applicable.

**Data Availability Statement:** Not applicable.

**Conflicts of Interest:** The authors declare no conflict of interest.

## Abbreviations

The following abbreviations are used in this manuscript:

| | |
|---|---|
| FDD | Fault Detection and Diagnosis |
| PCA | Prinicipal Component Analysis |
| DPCA | Dynamic Prinicipal Component Analysis |
| DNN | Deep Neural Network |
| NN | Neural Network |
| RNN | Recurrent Neural Network |
| TEP | Tennessee Eastman Process |
| LSTM | Long Short-Term Memory |
| GRU | Gated Recurrent Units |
| DLSTM-SAE NN | Deep LSTM Supervised Autoencoder Neural Network |
| DSAE-NN | Deep Supervised Autoencoder Neural Network |
| PRBS | Pseudo-Random Binary Signal |
| FDR | Fault Detection Rate |
| FAR | False Alarm Rate |
| GAN | Generative Adversarial Network |
| DSN | Deep Stacked Network |
| SAE | Stacked Autoencoder |
| OCSVM | One-Class SVM |
| SVM | Support Vector Machines |
| DL | Deep Learning |
| CNN | Convolutional Neural Network |

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
