# Peer review of "Hierarchical Deep LSTM for Fault Detection and Diagnosis for a Chemical Process"

_processes, doi:10.3390/pr10122557_

Round 1

Reviewer 1 Report

A hierarchical structure based on a Deep LSTM-SAE NN is presented for the detection and classification of faults in industrial plants. The proposed methodology has the ability to classify incipient faults that are difficult to detect and diagnose with traditional and many recent methods.

Thematically the work is interesting for the researchers and professionals and the proposed manuscript is relevant to the scope of the journal.

I found it appropriate for publication in the Processes journal, but only after some modifications and clarification from the Authors.

State specific contribution in the end on introduction section.

The presentation of the results should be improved. 

It would be appropriate to separate the Results and Discussion sections to improve the presentation of the results?

Define the research gap? How the proposed methodology covers the research gap.

The overall organization and structure of the manuscript are appropriate. The paper is well written and the topic is appropriate for the journal.
The aim of the paper is well described and the discussion was well approached, its results and discussion are correlated to the cited literature data.
In the introductory part, the authors give elaboration of the overall context stating the motivation and the objectives of the work, literature review of the research pathways .
The literature review is comprehensive and properly done.
The novelty of the work must be more clearly demonstrated.
The significance of the Work: Given the large number of analyzed data, this is an interesting study with a possible significant impact in this area.
Statistical interpretation of the analytical data must be more properly presented. The verification of the model should be performed. 

The work presented here is very interesting and well done, it is presented in a compact manner.

Reviewer 2 Report

The reported work in the manuscript is interesting and the authors have done a justifiable research. The quality of the work can be further improved by addressing the following queries.

1. Give the computational complexity analysis of the proposed algorithms.

2. The confusion matrix in figure 5 shows the classification of first level model with respect to 18 target class. But the first level model included 20 fault structures. Why 18 target class alone is considered?

3. Fault 9 and 15 are chosen for study with respect to injection of PRBS signal. State the reason behind your choice of fault.

4. The proposed method proved to be inefficient for fault 3, fault 10 and fault 19. How come the proposed process failed for these faults?
